# Protein Fishmeal Replacement in Aquaculture: A Systematic Review and Implications on Growth and Adoption Viability

**Edison D. Macusi** [1,2,*], **Melanie A. Cayacay** [2], **Elaine Q. Borazon** [3], **Anthony C. Sales** [4], **Ahasan Habib** [5], **Nur Fadli** [6] and **Mudjekeewis D. Santos** [7]

1. Faculty of Agriculture and Life Sciences, Davao Oriental State University (DOrSU), Mati City 8200, Davao Oriental, Philippines
2. Milkfish Assessment Project, Davao Oriental State University (DOrSU), Mati City 8200, Davao Oriental, Philippines; melaniecayacay1998@gmail.com
3. International Graduate Program of Education and Human Development (IGPEHD), College of Social Sciences, National Sun Yat-sen University, Kaohsiung 804, Taiwan; elaineqborazon@mail.nsysu.edu.tw
4. Department of Science and Technology (DOST-XI) Region 11, Davao City 8000, Philippines; dr.acs_dostxi@yahoo.com
5. Faculty of Fisheries and Food Science, Universiti Malaysia Terengganu, Kuala Nerus 21030, Terengganu, Malaysia; a.habib@umt.edu.my
6. Faculty of Marine and Fisheries, Universitas Syiah Kuala, Banda Aceh 23111, Indonesia; nurfadli@unsyiah.ac.id
7. National Fisheries Research and Development Institute (NFRDI), Bureau of Fisheries and Aquatic Resources (BFAR), Department of Agriculture, Quezon City 1100, Philippines; mudjiesantos@gmail.com
* Correspondence: edison.macusi@dorsu.edu.ph

**Abstract:** Aquaculture is growing rapidly as a food-producing sector and in recent years fishmeal prices have climbed more than two-fold on a global scale. This review of previous studies was performed to contribute to the extant literature on the aquaculture sector to aid cost reduction of aquafeeds by identifying substitute proteins that can replace fishmeal. The review followed the Preferred Reporting Items for Systematic Review and Meta-Analyses (PRISMA) using the SCOPUS and WOS (Web of Science), DOAJ (Directory of Open Access Journals), Academia, and PubMed Central databases. A total of 59 articles were included in the synthesis after screening for duplicates and articles that did not conform to the criteria. Results have shown that the 100% replacement of fishmeal with blood meal (BM) did not affect the growth of fish, nor did the 75% to 100% combination of poultry-by-product (PBM), feather meal (FEM), and BM. Moreover, a 10% replacement of fishmeal using seaweed (*Gracilaria arcuata*) had no adverse effect on the feed efficiency and growth performance of tilapia. Similarly, a 50% replacement of fishmeal using black soldier fly (*Hermetia illucens*), and a 25% replacement using soybean (*Glycine max*) also showed better results for fish growth. Our review shows that alternative protein can replace fishmeal in the aquaculture sector and reduce the cost of aquafeeds since alternative proteins are much cheaper than the usual fishmeal. Adoption of these alternative protein sources hinges on financial support, start-up incentives for companies, and ongoing studies on waste-to-feed production, which the government can also support.

**Keywords:** blood meal; feather meal; feeds; milkfish; tilapia

## 1. Introduction

Aquaculture is growing rapidly as a global food-producing sector and continues to flourish day by day [1–4]. It has been introduced in various regions of developing nations, including Africa and Asia, to provide rural communities with the chance to improve their quality of life and find a way out of poverty [5] by having a family income [6,7]. Fish meal represents 50% to 70% of the total material in fish feed [8]. It is highly considered a feed protein source since it has an excellent composition of amino acids and is easy to digest [9]. Moreover, the feed cost is 60% to 70% of an aquaculture farm's total operating

expenses. In contrast, the decreasing fish catches [10] and consistent growth in fishmeal consumption is creating a gloomy future for the aquaculture industry, although not if there is a paradigm shift toward the utilization of non-fish components for fish feed production. Over the next 20 years, aquaculture is projected to expand with increased demand for aquatic products, leading to higher fishmeal consumption. Fish oil (FO) and fishmeal (FM) could increase pressure on the diminishing stocks of marine fish resources [11,12]. In recent years, fishmeal prices have climbed globally by more than two-fold [13–15]. In Asia alone, fishmeal consumption for Nile tilapia climbed from 0.8 million tons to 1.7 million tons during the same period, while fish feed output increased from 40% in 2000 to 60% in 2008 [16]. According to the FAO [17], lowering the amount of fishmeal and fish oil in feeds is a huge step toward mitigating the pressure on global marine resource scarcity. Substituting fishmeal at the farm level could reduce expenses associated with production [18]. Moreover, combining a number of alternative protein sources with different limiting amino acids, such as lysine, methionine, threonine, and tryptophan has been strongly recommended [19]. The essential amino acid compositions of alternative protein sources for fish are not comparable to fishmeal. Fish meal (FM) is obviously inadequate to support the huge demand for fish feed with rising aquaculture production and use of feed in the industry placing a strong demand on fishmeal that cannot be sustained [20,21]. Therefore, there is an urgent need to find an alternative protein source to replace fishmeal [20,22]. Several sources of plant protein, single-cell protein, and animal protein have partially or entirely replaced the more expensive fishmeal. Animal protein sources have traditionally been regarded as the best alternative to replace fishmeal in the formulation of fish meals owing to their higher protein and fat content, superior essential amino acids, and excellent palatability [23]. On the other hand, plant ingredients that contain high protein content, high digestibility of crude protein, and low antinutritional components can replace fishmeal as a substitute protein source for fish [24]. Plant proteins are almost similar to fishmeal in protein content and amino acid digestibility. However, their amino acid profile does not match the amino acid requirements of some fish species, as fishmeal does. For example, methionine is the limiting amino acid in soybean meal (SBM), while corn gluten meal is deficient in lysine. Wheat gluten meal is limited in lysine and arginine [25]. According to De Francesco et al. [18], the impact of plant protein as a partial replacement for fishmeal shows contrasting results on the chemical composition of muscles. Soy products, including soybean meal and soy protein concentrate (SPC), have been researched as potential protein substitutes for fishmeal [26]. It has been frequently utilized as the most efficient alternative for fishmeal in aquaculture diets due to its high digestibility, high protein content, well-balanced amino acid composition, low cost, and consistent supply [27]. In addition, soybean concentrate can replace fishmeal for up to 40% to 100% [24]. Cocoa bean shell was reported to contain 13.2% to 17.7% crude protein and 13.0% to 16.1% fiber [28]. However, theobromine content of 1.3% in cocoa shell limits its usage in feeding, which is a downside in using it as a replacement for fishmeal. Moreover, seaweeds are a plant protein used to replace fishmeal in fish feed. They contain essential minerals, vitamins, pigments, compounds, fatty acids, and amino acids, which are highly required components for making fish feed [29]. Studies have shown that replacing some fishmeal with seaweed can improve growth, feed utilization, body composition, and disease resistance in fish [30]. Certain seaweed species, such as *Palmaria palmata*, have high levels of methionine, while seaweeds, such as *Laminaria digitata*, have lower amounts of methionine. Another protein used as a replacement for fishmeal is cacao pod husk (CPH); it is a by-product of cacao production in many tropical countries with cocoa production e.g. Southeast Asia and parts of Africa. Using this product as a replacement for fishmeal will eliminate environmental waste since it can be obtained at little to no cost to aquaculture farmers [31]. An early study on the growth of Nile tilapia (*Oreochromis niloticus*) fingerlings fed various levels of cocoa pod husk diets from zero (control), to 10% and 20% incorporation in the diet discovered gains in fish weight and specific growth rates to be higher with above 10% inclusion level in the diet [32]. Although this is the case, cocoa pod husk also has antinutritional factors such as theobromine; however, according to a

study by Ocran [33], the negative effect of antinutritional factors can be eliminated through fermentation. Following the study of Ogello et al. 2014 [25], the main terrestrial by-product meals used as a replacement for fishmeal are blood, insect, feather, and meat and bone. Regardless of its high crude protein content, these alternative proteins commonly lack amino acids, which limits the growth of the aquaculture species. One of the additional animal protein sources that can be utilized to replace fishmeal is poultry by-products. It was thought to be a significant replacement for fishmeal, particularly in rainbow trout since it has a similar composition of amino acids to fishmeal (FM) [34]. On the other hand, maggots are usually considered not to have any economic value. However, according to Ajani et al. [35], they have the potential to be a good source of animal protein in fish diets. Adesulu and Mustapha [36] also reported that some essential amino acids, including cystine, histidine, phenylalanine, tryptophan, and tyrosine are present in maggot meal and are higher than in fishmeal and soybean meal. Utilizing maggot meal as a source of protein for a fish diet is a good way of reducing the cost of waste disposal in the poultry industry, thereby helping to generate additional income for the fish and poultry industries. This review paper was performed to help the aquaculture sector reduce the cost of aquafeeds by identifying substitute protein sources that can be used in place of fishmeal and to assess the progress in feed development as an alternative to existing commercial feeds in the aquaculture industry.

## 2. Methodology

This review paper followed the Preferred Reporting Items for Systematic Reviews and Meta-Analyses (PRISMA) to address and evaluate the objectives of the study [37], PRISMA checklist, available in the Supplementary Material. To further understand the possible alternative protein sources for fishmeal and to assess its effect on the growth of cultured species, a comprehensive literature review using the same methods as in an earlier paper, i.e., [38,39], was conducted. The literature search in this review was limited from the year 2000 to the present, using 4 citation databases: Scopus/WOS, DOAJ (Directory of Open Access Journals), Academia, and PubMed Central. Data searching was first used to locate the records, and then, duplicates were eliminated. The articles that did not match the eligibility requirements were removed throughout the subsequent screening and data extraction processes. By looking through abstracts or contents, the remaining articles were evaluated to check their relevance to the topic of interest and whether they met the qualifying requirements. The reviewed literature was chosen based on inclusion criteria in the final phase based on the publications that passed the eligibility evaluation. The terms "fishmeal," "alternative protein source," and "aquafeeds" were used to ensure the search focused only on the literature related to the study.

Exclusion and inclusion criteria were applied to screen the articles used in this review. A total of 5331 articles were found in the Scopus database by using the keywords above (see Figure 1), which was reduced to 1714 by eliminating the duplicate articles found in the database. After removing the duplicates, articles were again screened to 288 by removing the articles that did not match the criteria for inclusion based on the title of the articles. Following the inclusion of criteria based on the abstract, 162 articles were record screened from the Scopus/WOS database. The same method was used on the open-access databases: DOAJ, PubMed, and Academia. A total of 124 articles were found in the 3 databases; 66 duplicate articles were removed, meaning the remaining 58 articles were recorded and screened based on the inclusion criteria. A total of 59 articles were included in the review on the basis that they passed the eligibility assessment.

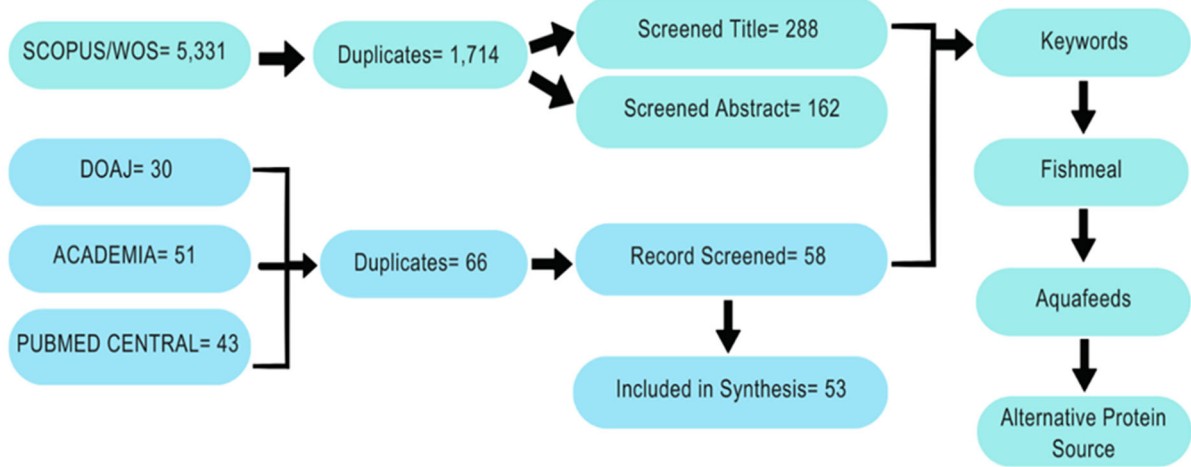

**Figure 1.** Flow of information using PRISMA.

## 3. Results

Figure 2 shows the distribution of the scientific production of fishmeal for the aquaculture industry: Egypt (18%), Brazil (12%), China (9%), Malaysia (6%), Thailand (5%), and the USA (5%) represented the top of the list. While Figure 3 presents the co-occurrence map of authors' keywords, while Figure 4 presents the co-occurrence map for titles and abstracts. A total of 49 author and index keywords and 44 text data from the articles' abstracts and titles were extracted and visualized in the co-occurrence map.

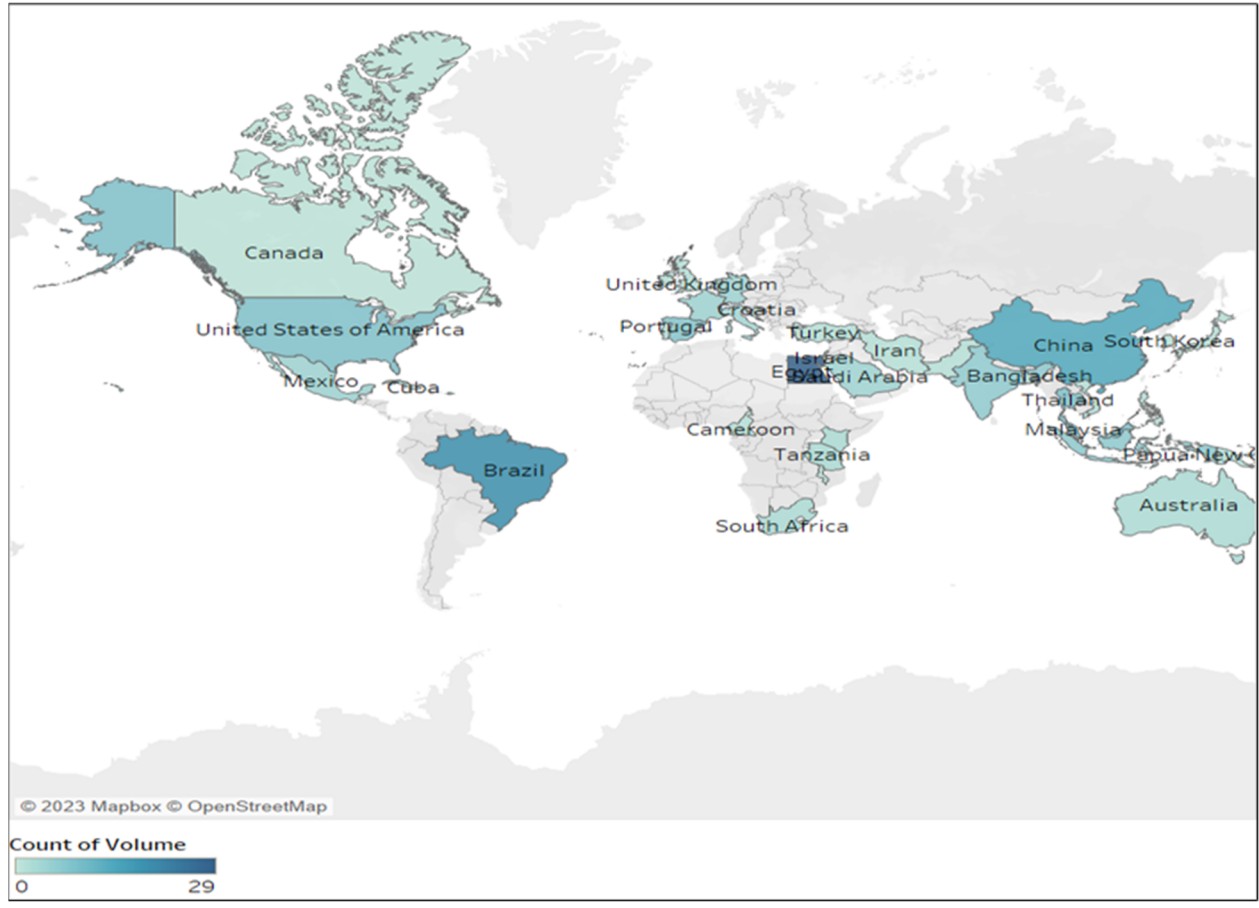

**Figure 2.** Distribution of global scientific production, with Egypt, Brazil, China, Malaysia, Thailand, and the USA at the top of the list.

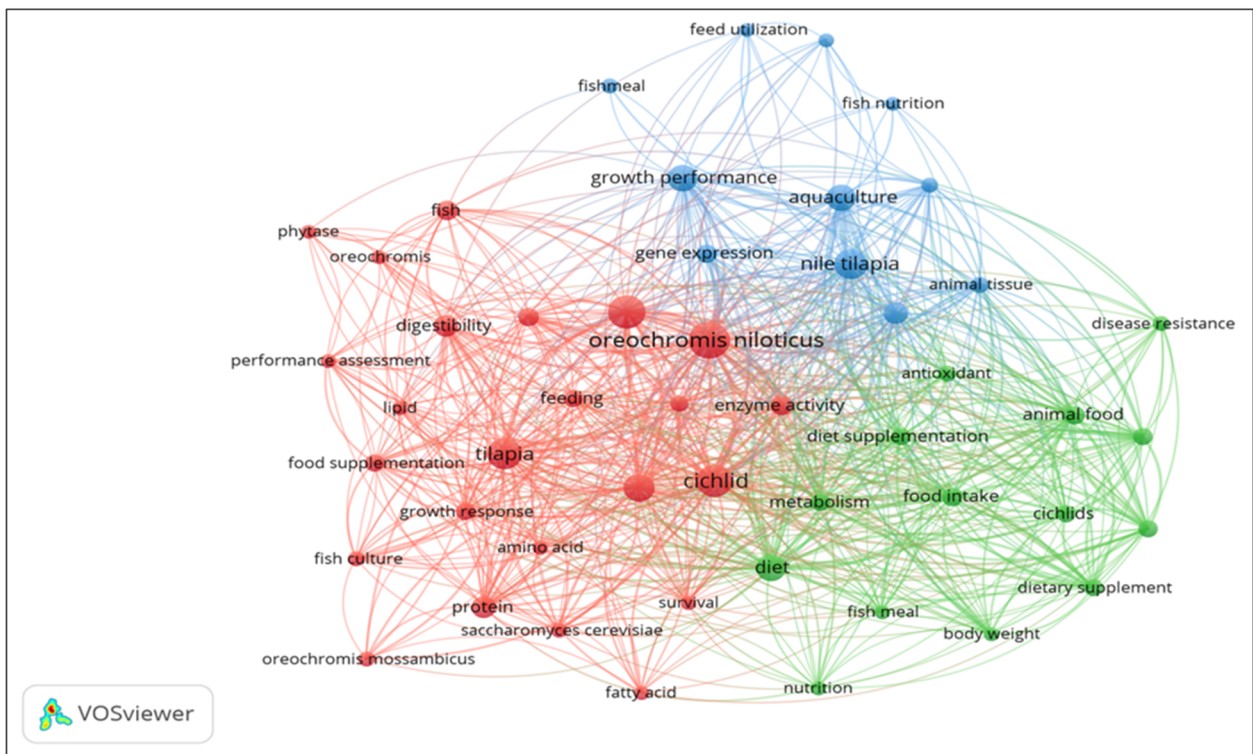

**Figure 3.** Map of connected networks based on keywords.

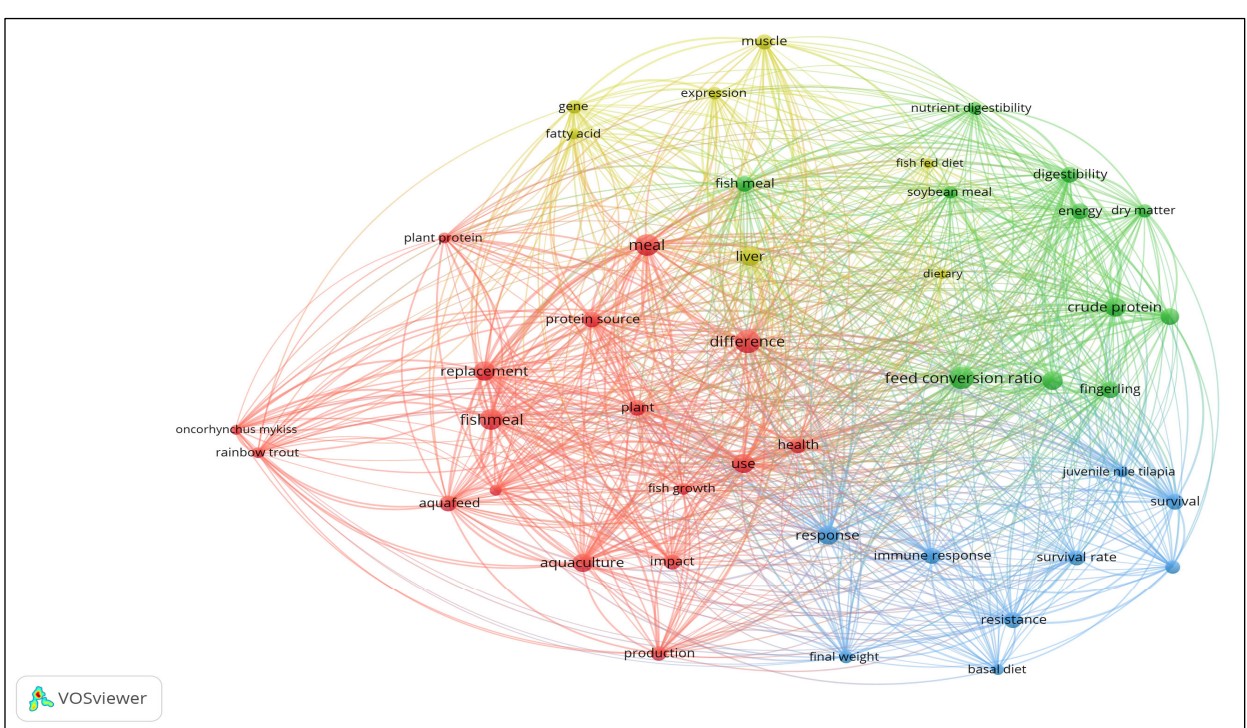

**Figure 4.** Map of connected networks based on titles and abstracts.

Table 1 presents the co-occurrence classification for the index and author keywords in terms of links, total link strengths, and occurrences. '*Oreochromis niloticus*,' 'growth, 'Cichlid,' 'tilapia,' and 'Nile tilapia' are keywords with the highest occurrences. In contrast 'Cichlid,' '*Oreochromis niloticus*,' 'diet,' 'growth rate,' and 'growth' are the keywords with the highest total link strengths. The cluster analysis of keywords shows three clusters, as

presented in Table 1. Under cluster 1, Cichlid, growth, growth rate, and *Oreochromis niloticus* have the highest link strength referring to the cultured Nile tilapia (*Oreochromis niloticus*). Under cluster 2, animal feed, animal food, diet, and food intake have the highest total link strength, which refers to the feed diet. For cluster 3, aquaculture, growth performance, immune response, and Nile tilapia have the highest total link strength, which refers to the effect of the diet on the growth of tilapia or cultured species.

**Table 1.** Occurrence classifications of texts from keywords (four keywords from the three clusters with the highest total link strengths are in bold).

| Keywords | Links | Total Link Strength | Occurrences |
|---|---|---|---|
| **Cluster 1** | | | |
| Amino acid | 22 | 36 | 5 |
| Artificial diet | 33 | 77 | 10 |
| Bacterium | 33 | 72 | 7 |
| **Cichlid** | 46 | **272** | 34 |
| Digestibility | 29 | 75 | 14 |
| Enzyme activity | 34 | 96 | 12 |
| Fatty acid | 17 | 27 | 5 |
| Feeding | 32 | 55 | 9 |
| Fish | 24 | 39 | 11 |
| Fish culture | 21 | 39 | 6 |
| Food supplementation | 27 | 57 | 8 |
| **Growth** | 44 | **179** | 39 |
| **Growth rate** | 46 | **188** | 25 |
| Growth response | 32 | 67 | 8 |
| Lipid | 21 | 33 | 5 |
| *Oreochromis* | 14 | 22 | 5 |
| *Oreochromis mossambicus* | 11 | 26 | 6 |
| ***Oreochromis niloticus*** | 45 | **264** | 47 |
| Performance assessment | 22 | 33 | 5 |
| Phytase | 11 | 14 | 5 |
| Protein | 34 | 96 | 14 |
| *Saccharomyces cerevisiae* | 26 | 48 | 6 |
| Survival | 28 | 44 | 5 |
| Tilapia | 42 | 159 | 31 |
| **Cluster 2** | | | |
| **Animal feed** | 31 | **101** | 8 |
| **Animal food** | 32 | **118** | 10 |
| Antioxidant | 25 | 48 | 7 |
| Body weight | 27 | 60 | 5 |
| Cichlids | 30 | 99 | 9 |
| **Diet** | 41 | **199** | 23 |
| Diet supplementation | 36 | 95 | 9 |
| Dietary supplement | 29 | 95 | 8 |
| Dietary supplements | 29 | 95 | 9 |
| Disease resistance | 18 | 32 | 6 |
| Fish meal | 26 | 42 | 5 |
| **Food intake** | 35 | **109** | 10 |
| Metabolism | 36 | 95 | 9 |
| Nutrition | 21 | 30 | 5 |
| **Cluster 3** | | | |
| Animal tissue | 29 | 66 | 7 |
| **Aquaculture** | 36 | **119** | 22 |
| Feed utilization | 10 | 14 | 5 |
| Fish nutrition | 8 | 10 | 5 |
| Fishmeal | 4 | 6 | 6 |

**Table 1.** *Cont.*

| Keywords | Links | Total Link Strength | Occurrences |
|---|---|---|---|
| Gene expression | 28 | 48 | 10 |
| **Growth performance** | 39 | **91** | 23 |
| Hematology | 11 | 17 | 5 |
| Histopathology | 29 | 57 | 6 |
| **Immune response** | 36 | **96** | 14 |
| **Nile tilapia** | 38 | **156** | 30 |

In terms of abstract and keywords (see Figure 4), the most common words based on the cluster analysis revealed four clusters. Words with the highest occurrences included 'difference,' 'feed conversion ratio,' 'meal,' 'fish meal,' 'response,' and 'crude protein.'

Table 2 presents the occurrence classification of texts from abstracts and titles in terms of links, total link strength, and occurrences. Under cluster 1, words with the highest total link strength included meal, fishmeal, difference, and replacement, which refer to the feed or diet used in the study. Under cluster 2, crude protein, digestibility, feed conversion ratio, and fish meal have the highest total link strength, which refers to the performance of the diet or feeds used. For cluster 3, immune response, resistance, response, and survival have the highest total link strength, again, in reference to the effect of the diet on the growth performance of the fish. Under cluster 4, the liver, gene, muscle, and expression have the highest total link strengths, which refer to the main organ examined in fish to check diet toxicity and immune response.

**Table 2.** Occurrence classifications of texts from abstracts and titles (four keywords from the three clusters with the highest total link strengths are in bold).

| Abstract/Title | Links | Total Link Strength | Occurrences |
|---|---|---|---|
| **Cluster 1** | | | |
| Alternative protein source | 35 | 98 | 12 |
| Aquaculture | 41 | 191 | 28 |
| Aquafeed | 39 | 155 | 21 |
| **Difference** | 43 | **224** | 42 |
| Fish growth | 39 | 83 | 13 |
| **Fish meal** | 42 | **250** | 34 |
| Health | 42 | 128 | 20 |
| Impact | 40 | 130 | 19 |
| **Meal** | 43 | **257** | 38 |
| *Oncorhynchus mykiss* | 27 | 77 | 10 |
| Plant | 41 | 142 | 22 |
| Plant protein | 32 | 86 | 11 |
| Production | 35 | 105 | 18 |
| Protein source | 41 | 157 | 22 |
| Rainbow trout | 27 | 83 | 11 |
| **Replacement** | 41 | **216** | 30 |
| Use | 43 | 195 | 29 |
| **Cluster 2** | | | |
| Body composition | 37 | 114 | 23 |
| **Crude protein** | 37 | **169** | 31 |
| **Digestibility** | 38 | **133** | 21 |
| Dry matter | 37 | 109 | 16 |
| Energy | 41 | 127 | 22 |
| **Feed conversion ratio** | 38 | **211** | 39 |
| Fingerling | 38 | 115 | 24 |

**Table 2.** *Cont.*

| Abstract/Title | Links | Total Link Strength | Occurrences |
|---|---|---|---|
| **Fish meal** | 43 | **154** | 21 |
| Nutrient digestibility | 37 | 87 | 15 |
| Protein efficiency ratio | 38 | 147 | 27 |
| Soybean meal | 37 | 84 | 14 |
| **Cluster 3** | | | |
| Basal diet | 31 | 60 | 12 |
| Dietary supplementation | 33 | 82 | 17 |
| Final weight | 36 | 88 | 15 |
| **Immune response** | 40 | **125** | 21 |
| Juvenile Nile tilapia | 34 | 66 | 12 |
| **Resistance** | 38 | **114** | 22 |
| **Response** | 41 | **192** | 31 |
| **Survival** | 36 | **119** | 22 |
| Survival rate | 33 | 83 | 19 |
| **Cluster 4** | | | |
| Dietary | 29 | 53 | 10 |
| **Expression** | 32 | **81** | 14 |
| Fatty acid | 28 | 64 | 12 |
| Fish fed diet | 32 | 48 | 10 |
| **Gene** | 34 | **111** | 17 |
| **Liver** | 42 | **154** | 29 |
| **Muscle** | 33 | **103** | 19 |

Figures 5 and 6 show an overlay visualization of the frequently used terms from the fishmeal studies' keywords, titles, and abstracts from 2000 to 2022. These present the trend in fishmeal research across the globe during the period. The recent studies are quite varied and include growth performance, gene expression, disease resistance, dietary supplement, replacement, resistance, response, health, and survival.

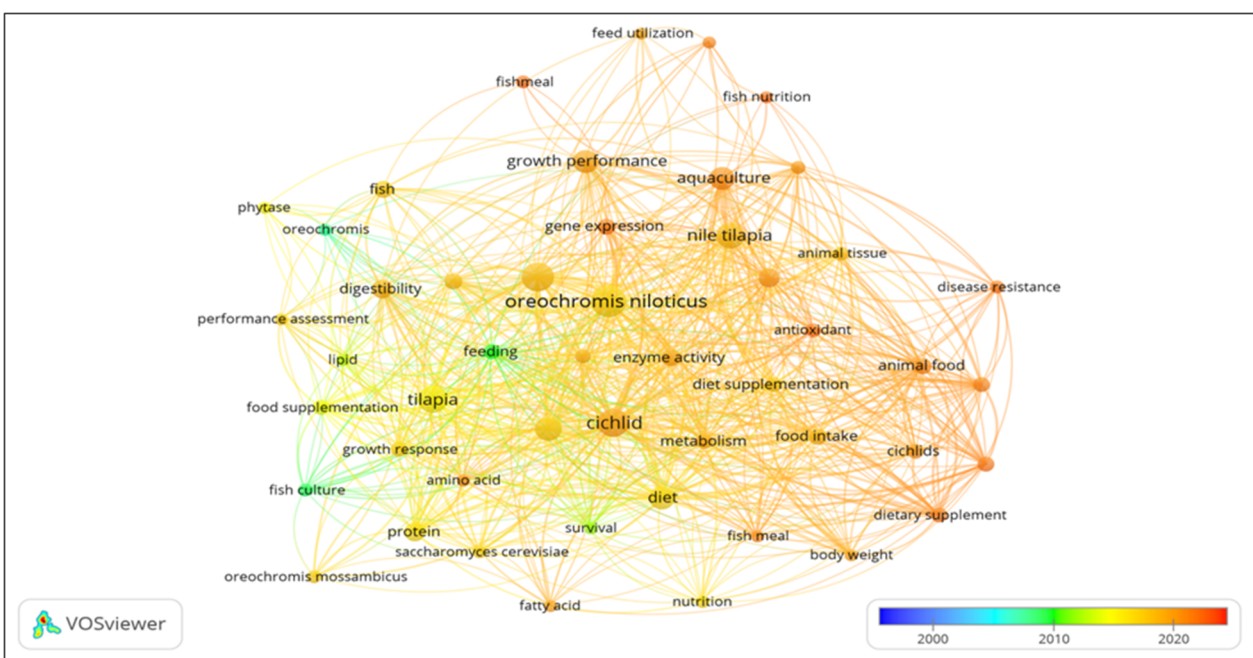

**Figure 5.** Overlay visualization of the most frequently used keywords in fishmeal studies from 2000 to 2022.

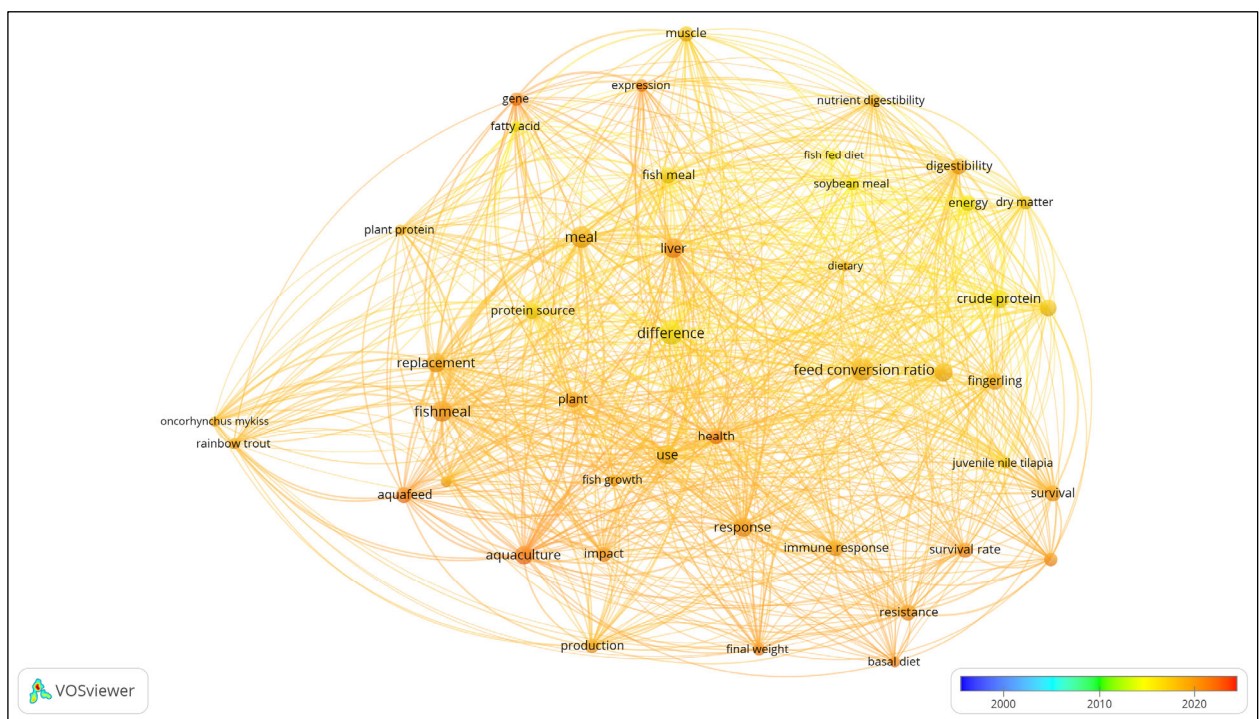

**Figure 6.** Overlay visualization of the most frequently used terms from abstracts and titles in fishmeal studies from 2000 to 2022.

Table 3 below summarizes the alternative proteins that can be used as a fishmeal replacement in aquafeed for culturing fish. From these replacements, blood meal can 100% replace the fishmeal and provide positive growth to the fish. In contrast, poultry-based by-products, feather meal, and bone meal can replace it with 75–100% efficiency, similar to the fishmeal diet. Seaweed can replace about 10% of the fishmeal diet in tests, while soybean meal can replace about 25% of the fishmeal diet, and insect-based diets can replace 50% of the fishmeal diet.

**Table 3.** Summary of different alternative protein sources and their effects on the growth of cultured fish.

| Protein Source | Percentage Replacement (%) | Growth Effect on Fish | References |
|---|---|---|---|
| Blood meal | 100 | Positive, increased the growth of fish | Aladetuhon and Sogbesan, 2013 [11] |
| PBM, FEM, and BM | 75 to 100 | Similar specific growth rate and weight gain compared to the control-diet-fed fish | Lu, Haga and Sato, 2015 [40] |
| Seaweed (*Gracilaria arcuata*) | 10 | No negative effect on feed efficiency and growth performance on Nile tilapia | Silva et al., 2015 [41] and Al-asgah, 2016 [42] |
| Black soldier fly (*Hermetia illucens*) | 50 | Better results for growth, protein utilization, and digestive functions | Melencion et al., 2022 [43] |
| Soybean meal (*Glycine max*) | 25 | No significant effect on tilapia's growth | Liu et al., 2017 [44] |

Legend: PBM (poultry by-product); FEM (feather meal); BM (bone meal).

## 4. Discussion

### 4.1. Effects of Substitute Proteins in Fishmeal on the Growth of Fish

From the result of this review, a variety of alternative proteins have been used to re-place fishmeal in fish diets, including animal and plant-based meals. Growth performance,

as determined by final weight and specific growth rate, revealed that extra protein could not be utilized efficiently for growth because growth energy was required to deaminate and extract ingested excess amino acids [45]. In the study conducted by Zlaugotne et al. [46], feed ingredients impacted fish and the environment, meaning it was necessary to evaluate feed materials and whether an alternative is possible that would be more effective and have less impact on the environment. The alternative fish feed must have high nutritional content, quality, high protein content, adequate amino acids, and digestibility and palatability. In addition, alternative fish feed should have insoluble carbohydrates, fiber, and low levels of heavy metals since these affect the fish growth process, alongside a low feed conversion ratio; feed costs must be economically justified as well, including the cost of feed production [47].

### 4.2. Animal Meal Replacement and Their Contribution to Fish Growth

Previous research conducted by Aladetuhon and Sogbesan [11] stated that adding blood meal to the experimental diet promoted the growth of fish from the start of the trial, the specific growth rate (SGR), weight gain (WG), biweekly growth rate (BGR), protein retention (PR), and food conversion ratio (FCR) have gradually increased and at its peak was 100% with the blood meal replacement. It is said that the survival rate was high in all the diets when using blood meal as a dietary supplement for fishmeal, which reveals a similar result to the study by Debbarma et al. [48], whereby blood-based meals could completely replace fish meal without affecting the fish's growth, survival, or the efficient conversion of feed in *Clarias gariepinus* (African catfish) fingerlings. In addition, previous research has discovered that fish fed with replaced fishmeal up to 75% to 100%, in combination with poultry by-product meal (PBM), feather meal (FEM), and blood meal (BM), demonstrated a better or similar specific growth rate and weight gain compared to the fish fed the control diet [40]. Moreover, no negative impact was observed on the final body weight, weight gain, or specific growth rate of rainbow trout (*Oncorhynchus mykiss*). It was reported that the rainbow trout grew better when fed with a mixture of poultry by-product meal, feather meal, and blood meal. Furthermore, insect meal can also be used as a protein replacement for fishmeal and is highly considered to be one of the most interesting protein substitutes. Fish feed production using insects can be one of the most environmentally and economically favorable methods [49]. According to a study by Melechon et al. [43], replacing 50% of the fishmeal with different larvae from insect meal from black soldier flies (*Hermetia illucens*) and mealworm (*Tenebrio molitor*) appears to have better results for growth, protein utilization, and digestive functions. In addition, liver histology and intermediary metabolism did not show any relevant changes, which was supported by intestinal histological differences between insect meals. In contrast, regardless of the limitations on the use of insects, terrestrial by-products, and fishery by-products as replacements for fishmeal, these animal protein sources have shown positive effects on feed conversion ratio, specific growth rate, final weight, and survival of different fish species in differently sized groups. However, in order for a transition to sustainable aquaculture to happen, it is essential that customers embrace the innovation and that the acceptable price and price thresholds of fish fed with alternative proteins, such as insect meal, are explored [50].

### 4.3. Plant-Based Meal Replacement and Their Input on Fish Growth

Soybean meal is considered one of the most suitable and reliable sources of alternative components for substituting fishmeal in commercial diets [27]. During a 7-week feeding trial, 25% of the dietary protein from a fish meal was substituted with soybean meal without significantly affecting the tilapia's ability to thrive [24]. In contrast, high levels of soybean meal (40–60% for juvenile fish) caused a reduction in growth and survival rates [44]. Similarly, the totoaba fish (*Totoaba macdonaldi*) can only tolerate up to 34.17% of soy protein concentration (SPC) substitution before the fish starts to develop some adverse effects, mainly due to the nondigestible carbohydrates and enzyme inhibitors

present in the soybean meal and soy protein concentrate [51]. On the other hand, the study by Silva et al. [41] utilized seaweed as a protein feed replacement for fishmeal and demonstrated that an inclusion level of up to 10% in practical diets had no adverse effect on feed efficiency and growth performance of Nile tilapia (*Oreochromis niloticus*). These results also confirm data obtained by Al-Asgah [42], who found that increasing the incorporation of red seaweed (*Gracilaria arcuata*), by up to 10%, had no adverse effect on the growth of African catfish (*Clarias gariepinus*). In contrast, the high inclusion level of red seaweed (*Gracilaria arcuata*), up to 20% to 30% in the diets of African catfish resulted in poor growth performance, feed utilization, and feed intake. However, future studies have recommended identifying a fishmeal replacement with no limitations and assessing the suitability of readily available alternative proteins as fishmeal replacements [45]. Feeding farmed fish with alternative protein is a viable solution to totally replacing fishmeal, although growth appears to be marginally affected by total replacement, yet only minimally enough to be economically practical. Therefore, it is recommended to continue expanding the knowledge for more sustainable and environmentally friendly aquaculture [38,52].

### 4.4. Alternative Protein Sources Benefit, Implications for Food Security, and Their Limits

The most pressing problem facing the aquaculture industry remains the feed cost, and there is considerable pressure on feed companies to develop less expensive formulations that maintain efficient growth at a lower cost per unit gain, to have less implications on environmental impacts [52,53]. To meet this goal, feed companies should lower the fishmeal levels further. Replacing the usual fishmeal with alternative protein sources can significantly benefit the fishing industry since these protein sources are far less expensive than fishmeal. On the other hand, alternative protein allows flexibility in feed formulations when feed ingredients fluctuate, which can also benefit the fishing industry greatly. According to Mulumpwa [54], for a fish product to be adequately available on the market might depend upon how alternative proteins are incorporated into fish feeds. Using alternative proteins as a replacement for fishmeal has the potential to increase fish production and improve food security. However, challenges remain to be resolved, such as food acceptance, food safety issues, and legislation, which can be dealt with by properly coordinating with governments– for instance, the Department of Agriculture, through the Bureau of Fisheries and Aquatic Resources (BFAR), and the fishing industry. Moreover, the lack of support, mainly by financial grants from the government could jeopardize the adoption of alternative proteins as a replacement for fishmeal.

### 4.5. Adoption Viability of Alternative Protein in the Aquaculture Industry

The spread of aquaculture production and intensification requires the search for high-quality, new efficient feed ingredients with low costs and sustainable production [55]. Fishmeal, the most expensive component in aquatic diets, is considered one of the most critical challenges in the development of the aquaculture industry. Given the rise in aquaculture production (e.g., tilapia, shrimps, and milkfish), and therefore, in aquafeed demand, replacing fishmeal with alternative protein sources will considerably reduce our dependence on fishmeal [56]. Significant gains in aquaculture production to supply additional protein, especially for freshwater fish, may also be made by combining alternative proteins or plant-based meals and animal-based meals to meet the requirements for fish growth [57]. While thorough knowledge is essential to balance multiple species, these systems have the added advantages of nutrient bioremediation and practical consumer perception [58]. Given these difficulties, technological advancements offer a great potential to generate consistently high-quality alternative protein products with improved nutritional characteristics. Some protein sources, such as fish by-products and insect meals, are feasible and promising alternatives compared to conventional fishmeal. In order for the substitution of fishmeal with other components, functional substances that can be utilized as feed supplements should also be employed to balance the nutritional components in the feed. Furthermore, using multiple protein sources allows for flexibility in feed formulations



when ingredient prices fluctuate, as feed manufacturers often use cost as a determinant in selecting ingredients [59]. Therefore, developing and optimizing alternative protein sources for aquafeeds will ensure a socially and environmentally sustainable future for the aquaculture industry.

## 5. Summary and Conclusions

This review was performed to provide insights that can be used for crafting policies in the aquaculture sector to reduce the cost of aquafeeds by identifying alternative protein sources, which can supply the aquaculture industry. Our findings have shown that blood meal is the most effective replacement and can substitute fishmeal up to 100% to provide a positive growth effect on the fish. In contrast, by-products from poultry, feather meal, and bone meal can be used in place of fishmeal but only with an effectiveness of 75–100%. Moreover, the use of seaweeds can also replace 10% of the fishmeal diet, soybean meal can replace 25%, and insect-based diets can replace 50% of the fishmeal diet. Adoption of these alternative protein sources hinges on financial support, start-up incentives for companies, and ongoing studies on waste-to-feed production, which the government could lead to ensure that a viable alternative protein for fishmeal replacement could be realized and adopted by the industry.

**Supplementary Materials:** The following supporting information can be downloaded at: https://www.mdpi.com/article/10.3390/su151612500/s1.

**Author Contributions:** Conceptualization, E.D.M., M.A.C. and A.C.S.; methodology, E.D.M., E.Q.B., M.A.C., A.C.S., A.H., N.F. and M.D.S.; software, E.Q.B. and M.A.C.; validation, E.D.M., M.A.C., A.C.S., A.H., N.F. and M.D.S.; formal analysis, E.D.M., M.A.C., A.C.S. and E.Q.B.; investigation, E.D.M., M.A.C., E.Q.B., N.F., A.H. and M.D.S.; resources, E.D.M. and A.C.S.; data curation, E.D.M. and M.A.C.; writing—original draft preparation, M.A.C.; writing—review and editing, E.D.M., M.A.C., A.C.S., A.H., N.F. and M.D.S.; visualization, E.Q.B. and M.A.C.; supervision, E.D.M., A.C.S., N.F. and M.D.S.; project administration, E.D.M.; funding acquisition, E.D.M. All authors have read and agreed to the published version of the manuscript.

**Funding:** The first author received funding from the Department of Agriculture, Philippine Rural Development Project (DA-PRDP), and the Department of Science and Technology Region 11 (DOST-XI) with the study entitled: *Enhancing food security, social inclusion, and sustainability in the milkfish aquaculture through the use of indigenous raw materials as feed components*.

**Institutional Review Board Statement:** Not applicable.

**Informed Consent Statement:** Not applicable.

**Data Availability Statement:** The data can be requested from the authors.

**Conflicts of Interest:** The authors declare no conflict of interest.

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
