# Peer review of "Protein Fishmeal Replacement in Aquaculture: A Systematic Review and Implications on Growth and Adoption Viability"

_sustainability, doi:10.3390/su151612500_

Round 1

Reviewer 1 Report

The main issue in the present manuscript is the category in another mean, if it is categorized as a review authors should remove the methodology and also remove the titles results and discussion then include the titles under results and discussion after introduction.

The other alternative is to categorize the manuscript as a research paper.

1-     Remove aquaculture from the keywords.

2-     Add space before it in line 43.

3-     Adjust space before to in line 44.

4-     Line 47: replace "makes up between 50 and 70 percent" with "represents 50 to 70%".

5-     Line 55: replace "due to" with which increased the demand for.

6-     Oreochromis niloticus in line 110 should be italic

7-     Remove the excess "t" in line 149.

Author Response

Comments/Suggestions

Answer

Line number

Reviewer 1

The main issue in the present manuscript is the category in another mean, if it is categorized as a review authors should remove the methodology and also remove the titles results and discussion then include the titles under results and discussion after introduction.

The other alternative is to categorize the manuscript as a research paper.

We appreciate you taking time in reading this work and offering feedback and recommendations. However, we have read and published a review paper on the same journal that uses this IMRAD format (e.g. https://doi.org/10.3390/su14052977) and the manuscript was acceptable as a review paper in this journal.

1. Remove aquaculture from the keywords.

The aquaculture keyword was already removed by the authors. Thank you so much for your comment to make this manuscript better.

In line 63

2.. Add space before it in line 43.

Thank you for taking time in helping to improve the paper. The authors have already made the changes and removed the unnecessary space in line 79

In line 79

3. Adjust space before to in line 44.

This was reflected in the manuscript.

In line 79

4. Line 47: replace "makes up between 50 and 70 percent" with "represents 50 to 70%".

We have already changed to your preferred suggestion

In line 82

Reviewer 2 Report

What can you say about the availability of articles in the databases? Do you think other search criteria could improve the search for bibliographic material? What can you conclude from this in this regard?

Author Response

Comments/Suggestions

Answer

Line number

Reviewer report 2

1. What can you say about the availability of articles in the databases? Do you think other search criteria could improve the search for bibliographic material? What can you conclude from this in this regard?

The authors have gathered enough articles from all the databases. From SCOPUS/WOS down to Academia.edu, PubMed Central, and DOAJ (Directory of Open Access Journals). All the articles collected were screened on the basis of their titles, abstract, keywords and content. Though some articles were not available locally in the region, but the gathered articles were sufficient to help us create a review paper with regards to fishmeal protein replacement or substitution. Our study was limited to our objectives, although opening it to more search word criteria could expand its scope, this could actually go beyond the objectives of our study.

Reviewer 3 Report

Manuscript is not well designed and discussed. Some novel literatures are missing. For instance, if you do protein review paper, you should use at least one of his paper (https://scholar.google.com/citations?hl=tr&user=ChMzbnwAAAAJ&view_op=list_works&sortby=pubdate).

Figure 2 is low quality. I advice to not suitable for publish.

-

Author Response

Comments/Suggestions

Answer

Line number

Reviewer 3

1. Manuscript is not well designed and discussed. Some novel literatures are missing. For instance, if you do protein review paper, you should use at least one of his paper (https://scholar.google.com/citations?hl=tr&user=ChMzbnwAAAAJ&view_op=list_works&sortby=pubdate).

Figure 2 is low quality. I advice to not suitable for publish.

The authors value your comment on how to improve this paper. However, the authors obtained the articles in the synthesis from highly regarded database sources like SCOPUS and WOS, including DOAJ, PubMed Central, and Academia.edu. We would want to include your suggested paper as well, but we were unable to locate it in the database used for this study that was mined because the inclusive years of study were from 2000-2022. Thank you.

Reviewer 4 Report

The review of Macusi et al represents the world state of fishmeal replacers. This work includes high quality analysis and nice Meta-Analyses visualization. But the title and aim of the review is not fit the current content. In my point of view the authors have two choice:

-          Make focus on the situation in Philippines including previous studies and the local alternative proteins and the best local alternative and the most promising international alternatives, in this case the novelty of the review will be increased.

-          Remove Philippines from the title and aim to fit the current content, but in this case the novelty may be lost.

I hope if the authors can do such analysis for the previously published studies in Philippines and the surrounding area compared to the other world.

The conclusion did not give a clear message to readers or farmers.

The language need improvement and native proof reading.

Accordingly, the review need much efforts from the authors to be reconstructed and to have peace of art. 

The attached PDF contains several corrections that need to be considered.

The language need improvement and native proof reading.

Author Response

Comments/Suggestions

Answer

Line number

Reviewer report 4

1. The review of Macusi et al represents the world state of fishmeal replacers. This work includes high quality analysis and nice Meta-Analyses visualization. But the title and aim of the review is not fit the current content. In my point of view the authors have two choice:

-          Make focus on the situation in Philippines including previous studies and the local alternative proteins and the best local alternative and the most promising international alternatives, in this case the novelty of the review will be increased.

-      Remove Philippines from the title and aim to fit the current content, but in this case the novelty may be lost.

The authors appreciate your feedback on how to make this paper better. The authors have removed the Philippines, a specific country, from the title as suggested in order for it to suit the manuscript's existing content.

New title:

“Protein fishmeal replacement in aquaculture: A systematic review and implications on growth and adoption viability”

2. The conclusion did not give a clear message to readers or farmers.

In order to give readers a clear message that blood meal has the most significant percentage to substitute fishmeal with a good growth effect on the fish, the conclusion has been modified and expanded to highlight the study's significant findings. Moreover, the goal of the study is to inform the aquaculture sector that a possible alternative protein could help the aquaculture sector reduce the cost of fish feeds with an additional limiting factor that could hinder the adoption of alternative protein. Thank you for your time looking through this paper and making some comments to improve it.

In line 396 to 401

3. The language need improvement and native proof reading.

In order for readers to better comprehend, the authors have proofread this manuscript already. 

4. Accordingly, the review need much efforts from the authors to be reconstructed and to have peace of art. 

We appreciate the time you took to review this manuscript and offer and give suggestions for improvement. For this work to be improved, we have added and reflected the proposed comments. Thank you!

Author Response

Comments/Suggestions

Answer

Line number

1. add a space in line 43

Thank you for your comment we have added space in line 43.

2. remove double space in line 44

Double space was removed.

3. re-write this sentence, delete local, delete a means of in line 45

The authors have re-written the sentence and deleted the words you suggested to remove.

- It has been introduced in various regions of developing nations, including Africa and Asia, to provide local rural communities with the chance to improve their quality of life and find a way out of poverty (Olaganathan & Kar Mun, 2017) and by producing family income (Moisa et al., 2022: Tran et al., 2023).

In line 78 to 82

4. end the sentence and start new one in line 48

To improve this manuscript, the authors have incorporated your suggestions. Thank you.

- Fish meal represents 50 to 70 percent of the total material in fish feed (Jannathulla et al., 2019). It is highly considered a feed protein source since it has an excellent composition of amino acids and is easy to digest (Olsen & Hasan, 2012).

5. may the authors means increasing demand for aquatic products, which will lead to increase fishmeal consumption.

We appreciate your comment in order for this manuscript to improve. The authors have modified the sentence to make it clearer.

-        Over the next 20 years, aquaculture is projected to expand which  increased demand for aquatic products, which lead to increase fishmeal consumption.

6. please unify, fish meal or fishmeal

The authors have unified the word fishmeal in the manuscript. Thank you.

7. very long sentence. Please consider along the MS in line 65 to 69

To improve this manuscript, the authors have reflected your suggested comment. Thank you.

- Moreover, combining a number of alternative protein sources with different limiting amino acids like lysine, methionine, threonine, and tryptophan has been strongly recommended (Ogunji & Wirth, 2001). The essential amino acid compositions of alternative protein sources for fish are not comparable with that of fishmeal.

8. delete “ . “ in line 78

The authors have reflected your comment. Thank you.

9. rearrange this paragraph, delay the part of seaweeds after cocoa. Also divide this part for two in line 88-92

Thank you so much for reviewing this manuscript.

The authors have re-arranged the paragraph according to the suggested comment and divided it in to two parts.

- Soy products, including soybean meal and soy protein concentrate (SPC), have been researched as potential protein substitutes for fishmeal (Trejo‐Escamilla et al., 2017). It has been frequently utilized as the most efficient alternative for fishmeal in aquaculture diets due to its high digestibility, high protein content, well-balanced amino acid composition, low cost, and consistent supply. (Zhou et al., 2005).

10. itilized the latin name

The authors have italized the scientific names. Thank you.

11. remove “T” in line 149

The author has deleted the unnecessary letter from line 149. I appreciate you fixing the typographical mistake.

12. Discussion “Growth effect of alternative protein for fishmeal” needs to be re-write

Thank you for taking the time to comment and help this manuscript. The authors have reflected your comment on the paper.

- Effects of substitute proteins for fishmeal on the growth of fish

13. The conclusion must introduce a direct advice, which the pest plant or animal protein alternative. What is the best choice for Philippines.

We appreciate your comment on how to improve this manuscript. The conclusion has been rewritten to emphasize the findings of the study.

Round 2

Reviewer 1 Report

I would like to thank authors for their valuable effort in revising the present manuscript, now this work become more organized, the English language is acceptable, and all of my comments were addressed carefully

Reviewer 3 Report

-

it is fine.

Reviewer 4 Report

THE AUTHORS IMPROVED THE REVIEW SUBSTANTIALLY AND CONSIDER THE SUGGESTED COMMENTS.

Reviewer 5 Report

Thank you for improving the manuscript.

Minor grammatical check is required.